# Recent Advances in Polycaprolactones for Anticancer Drug Delivery

**DOI:** 10.3390/pharmaceutics15071977

**Published:** 2023-07-19

**Authors:** Abhi Bhadran, Tejas Shah, Godwin K. Babanyinah, Himanshu Polara, Somayeh Taslimy, Michael C. Biewer, Mihaela C. Stefan

**Affiliations:** Department of Chemistry and Biochemistry, The University of Texas at Dallas, Richardson, TX 75080, USA; abhi.bhadran@utdallas.edu (A.B.); tejas.shah@utdallas.edu (T.S.); godwin.babanyinah@utdallas.edu (G.K.B.); himanshu.polara@utdallas.edu (H.P.); sxt163130@utdallas.edu (S.T.); biewerm@utdallas.edu (M.C.B.)

**Keywords:** polycaprolactones, ring opening polymerization, catalyst, functional caprolactone monomers, drug delivery systems, polymeric micelles, targeting, stimuli-responsive polymers

## Abstract

Poly(ε-Caprolactone)s are biodegradable and biocompatible polyesters that have gained considerable attention for drug delivery applications due to their slow degradation and ease of functionalization. One of the significant advantages of polycaprolactone is its ability to attach various functionalities to its backbone, which is commonly accomplished through ring-opening polymerization (ROP) of functionalized caprolactone monomer. In this review, we aim to summarize some of the most recent advances in polycaprolactones and their potential application in drug delivery. We will discuss different types of polycaprolactone-based drug delivery systems and their behavior in response to different stimuli, their ability to target specific locations, morphology, as well as their drug loading and release capabilities.

## 1. Introduction

Aliphatic polyesters, including poly(ε-caprolactone) (PCL), poly(glycolide) (PGA), poly(lactide) (PLA), and Poly(lactide-co-glycolide) (PLGA), are commonly used for drug delivery applications owing to their biocompatibility and biodegradability [1,2,3]. These polyesters are approved by the US Food and Drug Administration (FDA) due to their ability to address several challenges that occur during the drug delivery process [4,5]. They demonstrate the potential to optimize pharmacokinetics by precisely controlling drug concentrations within the therapeutic range, leading to a reduction in adverse effects [4]. PCL, in particular, stands out due to its flexibility in introducing different substituents into its backbone. PCL has many therapeutic applications, such as tissue engineering scaffolds, artificial blood vessels, wound dressings, nerve regeneration devices, and drug delivery devices [6,7,8,9]. Due to its good mechanical properties and high permeability to many drugs, PCL is a great choice for controlled drug delivery applications [1,10]. One of the significant advantages of PCL over other aliphatic polyester is its ability to fine-tune the physical and chemical properties. This is achieved by functionalizing their monomers with the desired substituents. In this way, the properties of PCL can be altered to our advantage and can be applied to precisely control the loading and release of drug molecules from PCL-based drug delivery systems [1,4,11,12,13,14,15,16]. 

PCL is a semi-crystalline polymer with the ability to adjust its crystallinity based on its molecular weight [17,18]. When the molecular weight of PCL is increased, it results in longer chain lengths. Consequently, chain folding can occur, leading to a reduction in crystallinity [19]. PCL has a relatively lower melting point (60 °C) and lower glass transition temperature (−60 °C) when compared to other aliphatic polyesters [17,19,20,21]. The low intermolecular interactions and high mobility of chain segments in PCL are responsible for its very low melting and glass transition temperatures [17]. As a result, PCL shows elastic behavior at room temperature and makes it easy to transform into various structural forms [19]. PCL has gained popularity as a biomaterial due to its exceptional rheological and viscoelastic properties, making it a valuable material in tissue engineering [20]. It is widely used in the development of controlled-release contraceptives and matrix implants, where its slow degradation rate allows for long-term drug release and stable structural support [22]. Moreover, PCL is also considered a promising material for developing scaffolds that can mimic the structure and mechanical properties of cardiac tissues. Its desirable chemical, mechanical, and biocompatible properties and biodegradability make it a promising candidate for re-engineering and regenerating the myocardium after disease or injury [23].

## 2. Synthesis of Poly(ε-Caprolactone)s

PCL can be easily synthesized by the polycondensation of 6-hydroxycaproic acid. However, the most preferred technique for generating PCLs is through the ring-opening polymerization (ROP) of ԑ-caprolactone (CL) monomers (Figure 1) [21]. This method offers numerous advantages, including higher degrees of polymerization and molecular weight, superior product quality, and low-dispersity polymers [18,24]. Various mechanisms such as anionic, cationic, radical, and coordination processes can be employed for the ROP of PCLs [1,25]. Each approach significantly influences the resulting copolymers’ final molecular weight, end group composition, molecular weight distribution, and chemical structure [6,26]. However, the first three methods exhibit drawbacks, such as backbiting and low polymerization control. Consequently, a metal-based catalyst is employed to polymerize CL through a coordination-insertion process [18,27,28]. 

### 2.1. Ring-Opening Polymerization Using Metal Catalyst

In this polymerization, the presence of an alcohol initiator and a metal catalyst is necessary. Among the widely used tin-, aluminum-, and zinc-based metal catalysts, Tin (II) octanoate (Sn(Oct)_2_) is extensively utilized in the synthesis of PCLs as it was approved safe by the FDA and can produce high molecular weight PCLs with low polydispersity and reduced transesterification [18,27,28,29,30,31]. An overview of the mechanism of ROP of CL monomers using Sn(Oct)_2_ catalyst is shown in Figure 2. In this mechanism, the monomer is coordinated to the metal center, followed by the insertion into a metal-alkoxide species through the acyl-oxygen bond. 

However, Sn(Oct)_2_ is classified as toxic and found to be less effective for polymerizing functional CL monomers [32,33,34,35,36]. The functionalities present in the CL monomer can sometimes affect the polymerization kinetics, reducing the reaction rate. The use of Sn(Oct)_2_ often requires longer reaction times and higher temperatures to achieve a reasonable polymerization rate [35]. On the other hand, zinc undecylenate (ZU) is an inexpensive antifungal agent used in pharmaceutical applications and is also a great ready-to-use catalyst candidate. Zinc catalysts show low toxicity when compared to tin catalysts and have been shown to provide *living* polymerization of CL monomers. Zinc catalysts require no purification before use and can be easily removed after polymerization, making them a better choice than Sn(Oct)_2_ [37,38]. Stefan et al. used a ZU catalyst to polymerize γ-functionalized CL monomers to develop linear- and star-like block copolymers. The produced polymers showed monomodal distribution and were used to study the difference in their size, morphology, thermodynamic stability, and drug loading capacity [38]. 

### 2.2. Ring Opening Polymerization Using Rare-Earth Metal

Most ROP strategies have focused on polymerizing unsubstituted CL monomers rather than substituted ones. However, a catalytic system that can effectively polymerize both substituted and unsubstituted ε-CL monomers is desirable for practical applications. Rare earth metal catalysts have been gaining attention recently due to their high reactivity, nontoxic nature, and mild reaction conditions [36,39,40,41,42]. In particular, Neodymium (Nd) complexes containing ligands such as—alkoxides, phenolates, and iminophosphoranes have exhibited excellent catalytic activity in the ROP of cyclic esters [43,44,45]. Using this catalytic system, the transesterification of PCL can be reduced by the incorporation of bulky ligands into the metal center [46]. Stefan et al. used an Nd catalytic system to polymerize ester-functionalized CL monomers. Here, they studied the catalytic activity of a newly developed Nd-based catalytic system, NdCl_3_·3TEP/TIBA (TEP = triethyl phosphate, TIBA = triisobutylaluminum), to polymerize γ-4-phenylbutyrate-CL, a prodrug monomer and compared it with the traditional Sn(Oct)_2_ catalytic system. They observed a relatively higher conversion rate of the monomer leading to higher molecular weight and relatively lower transesterification when compared to the polymer synthesized with tin catalyst (Figure 3) [36].

### 2.3. Ring-Opening Polymerization Using Organic Catalyst

A significant challenge in conventional polymerization processes using organometallic catalysts is the intricate and expensive removal of residual metals from the resulting polyesters. This issue becomes particularly crucial in biomedical applications, as medical-grade polymers are required to adhere to strict standards [47]. As an alternative to metal catalysis, organocatalysis for ROP is also growing fast [48]. Organocatalytic ROP offers significant advantages, including fast reaction rates, generating polymers with narrow polydispersity, and performing the reaction under ambient temperature conditions [34]. Commonly used organocatalysts include 1,5,7-triazabicyclo[4.4.0]dec-5-ene (TBD), 1,8-diazabicycloundec7-ene (DBU), and phosphazene bases [49,50,51,52,53]. DBU exhibits high catalytic activity for ROP at ambient temperature. However, extended reaction times with DBU can sometimes lead to transesterification reactions. Moreover, DBU requires a cocatalyst to polymerize CL monomers, and the monomer conversions are relatively low [54,55]. In contrast, TBD and phosphazene bases display exceptional catalytic performance towards ROP conditions.

#### 2.3.1. Organocatalytic ROP Using Phosphazene Bases

Phosphazene bases are excellent catalysts towards ROP and can carry out polymerization at temperatures as low as −78 °C. However, they exhibit lower activity regarding the ROP of lactones [54,56]. Hadjichristidis et al. investigated the ROP of CL monomer using a phosphazene base (*t*-BuP_2_) in different solvents using various protic initiators. They observed that the dispersity of PCL became broader as the conversion of CL monomers increased, indicating that transesterification reactions occurred on the polyester chains in all the cases [56]. However, in their recent study, they compared the catalytic activity of different organocatalysts for the ROP of a CL derivative, N-Boc-1,4-oxazepan-7-one (OxPBoc). The study revealed that a binary organocatalytic system consisting of phosphazene/thiourea (*t*-Bu-P_4_/TU1) leads to a controlled/living ROP compared to DBU and TBD. As expected, DBU exhibits the lowest activity for ROP due to its relatively weak basicity. In contrast, ROP conducted under TBD demonstrates rapid polymerization and observed 95% conversion within 6 min. Nevertheless, the strong basicity of TBD hinders control over the molecular weight and dispersity of the resulting polymer compared to the *t*-Bu-P_4_/TU1 binary system [57].

#### 2.3.2. Organocatalytic ROP Using TBD

TBD is recognized as one of the highly efficient organocatalysts for the ROP of cyclic esters due to its strong basicity [58,59]. TBD as a catalyst for the ROP of CL monomers was first investigated by Hedrick, Waymouth, et al. and was shown to produce PCL with narrow dispersity [50,51]. TBD exhibits a bifunctional catalytic mechanism, facilitating the activation of the monomer and the initiator through hydrogen bonding interactions. TBD activates the carbonyl group of CL monomer by donating a hydrogen bond, making it more susceptible to nucleophilic attack by the alcohol. Simultaneously, TBD activates the alcohol by accepting a hydrogen bond to its imine-like nitrogen. TBD then facilitates the ring-opening process by hydrogen bonding to the former carbonyl oxygen in the tetrahedral intermediate and transferring the hydrogen from the alcohol to the ring oxygen adjacent to the former carbonyl group (Figure 4) [50].

TBD is known for its versatility in synthesizing functional PCL block copolymers, and when compared to the other amidine bases, TBD showed a relatively higher catalytic activity [32,34,50]. TBD-catalyzed ROP is reported to provide *living* polymers from amide-functionalized CL monomers more efficiently than Sn(Oct)_2_ catalysts. For example, Lang et al. studied the ROP kinetics of several amide-functionalized CL monomers utilizing TBD catalyst [34]. They compared their results with those obtained from γ-ester functionalized analogs. The study revealed that TBD-catalyzed ROP of amide substituents went through a controlled and living manner compared to the ester-functionalized CL monomer. This was further confirmed through density functional theory, which showed that the enthalpy of ring-opening of γ-amide CL was more negative than that of γ-ester CL.

## 3. Functional PCL

Functionalization of PCL is highly desirable as it can control the mechanical and thermal properties while influencing the kinetics of polymerization [1,60]. By introducing different functional groups to the polymer backbone, crystallinity, hydrophilicity, biodegradation rate, and bioadhesion can be fine-tuned [16,61]. Moreover, functionalized PCLs offer opportunities for modification by incorporating drugs, bioactive moieties, and stimuli-responsive components. Most reported functionalized CL monomers are γ-functionalized through an ether linkage due to ease of functionalization and less interference in ROP [15,62]. Stefan's group previously reported several γ-ether functionalized CL monomers and studied their application in drug delivery [11,12,13,37,63,64]. Subsequently, they also reported γ-ester functionalized CL monomers and investigated the possibility of ester-functionalized PCLs. Ester linkages in PCLs are promising as it provides access to many natural substituents through simple esterification reactions. Nonetheless, the presence of two ester groups promoted transesterification side reaction during ROP [36,65,66]. Some of the examples of functional CL monomers that were reported after 2015 are listed in Figure 5.

Functional PCL can be synthesized by two strategies: (i) polymerization of functional CL and (ii) post-polymerization chemical modification. In the first case, functional groups can be attached to different positions of CL monomer, with α- and γ-functionalities being the most observed. In contrast, the second case is mostly used for substituents not amenable to ROP conditions. Therefore, these substituents are directly attached to the side chain of PCL after polymerization. PCL-based polymers with pendant functional groups such as silane, alkoxide, halogen, azide, and unsaturation were reported [67,68,69].

**Figure 5 pharmaceutics-15-01977-f005:**
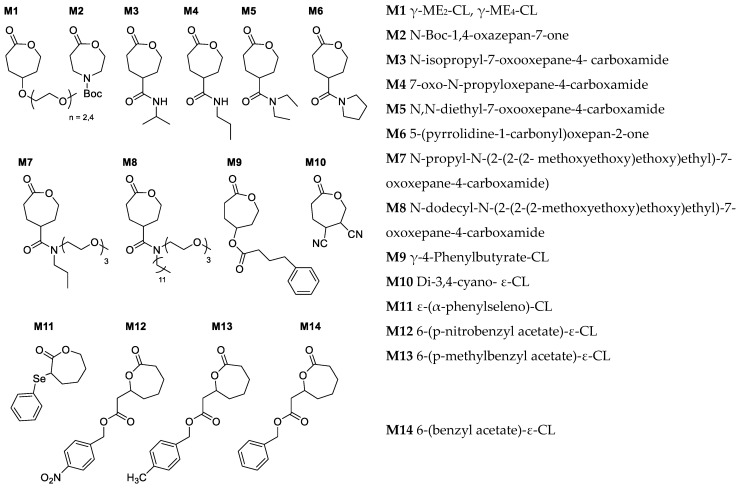
Examples of recently reported functional CL monomers [32,34,36,57,64,70,71,72,73].

### 3.1. Halogen Functionalized PCLs

As halogens are amenable to ROP conditions, halogen-functionalized PCLs can be easily synthesized by the ROP of halogen-functionalized CL monomers. Halogen-functionalized CL offers versatility in its applications, serving as both a monomer for ROP and an initiator for atom transfer radical polymerization (ATRP) [74,75]. Bexis et al. demonstrated the potential of this approach by synthesizing a range of bromine (Br) substituted PCL-based graft copolymers (Figure 6). They achieved this by utilizing Cu(0) as a catalyst to polymerize methyl acrylate (MA) with Poly(α-Br-CL) homopolymer and Poly(α-Br-CL-co-ε-CL) copolymers as macroinitiators. Notably, this process produced polymers with low dispersity (Đ ≤ 1.22) and unique macromolecular topologies and was conducted under mild reaction conditions. Employing poly halogen-functionalized ε-CLs as macroinitiators make it possible to generate living grafted polymers with a degradable poly(ε-CL) backbone in a controlled manner [76].

### 3.2. Propargyl Functionalized PCLs

The polymerization of certain functional CL monomers can sometimes be challenging due to the interference caused by the functional group under ROP conditions. In that case, the functional groups are often protected before polymerization to reduce the risk of unwanted side reactions and then deprotected after polymerization to generate the corresponding polyesters. Polar functional groups like hydroxyl, amino groups, and carboxylic acid are functionalized to PCL in this manner [62,77,78,79]. However, the protection method is not always suitable for PCL due to the harsh deprotection conditions that can sometimes degrade the ester backbone of the PCL. Functional groups that do not demand protection during ROP are available. These groups will not interfere with the polymerization mechanisms and can be modified to achieve the desired interaction with the cargo through post-polymerization reactions. For example, Emrick et al. introduced the propargyl group to the α-position of CL monomers to develop alkyne-functionalized aliphatic polyesters. They explored using orthogonal click reactions to enable the sequential functionalization of a diblock polyester [69,80]. The propargyl group can be a versatile platform for functionalizing biologically relevant substituents through post-polymerization modification via click chemistry. Propargyl-substituted polyesters were used in click chemistry to conjugate drugs, stimuli-responsive moieties, or targeting agents. Other biologically relevant molecules, such as drugs, peptides, and antibodies that are not tolerated in ROP, can be tagged in this way [69].

### 3.3. Amide Functionalized PCLs

Amide functionalized PCL, an area explored recently, has excellent potential to develop advanced materials for drug delivery applications with minimal effort. One of the significant advantages of amide linkage is that it can accommodate two functional groups, which doubles the grafting density of the functional group per monomer [32]. For example, Stefan et al. reported TBD-catalyzed ROP of γ-amide functionalized ε-CL monomers. Their work demonstrated that TBD could initiate the synthesis of amphiphilic homopolymers as well as amphiphilic block copolymers (Figure 7) [32,70].

### 3.4. Ester Functionalized PCLs

Jayakannan et al. incorporated the carboxylic acid group into PCL with the help of a protecting group. The butyl ester protecting group of the generated polymers was removed after polymerization to generate the carboxylic acid functionality. The produced polymer was hydrophilic in nature and was used to make nanoparticles for the delivery of Doxorubicin (DOX) [81,82]. Lang et al. developed a more controlled way of synthesizing carboxylic acid functionalized PCL by tuning the electronic effect and steric hindrance of the protecting group (Figure 8). The results revealed that the electron-donating group exhibited greater control over the degree of polymerization and enabled efficient removal of the protecting groups without causing any degradation to the polymer backbone [73].

## 4. PCL for Drug Delivery

Enhancing the efficiency of a drug is possible by designing biocompatible nanocarriers that allow the delivery of the drug in a controlled and less toxic manner [63]. Various drug delivery systems (DDS) have been employed, and most comprise synthetic polymers, including micelles, dendrimers, nanogels, and nano-capsules [83,84,85,86,87,88,89]. PCL are excellent candidates for drug delivery applications due to their biodegradability, biocompatibility, and synthetic versatility [90,91,92,93]. Effective utilization of PCL as a drug delivery carrier can be observed in the case of the Capronor implant, enabling an extended release of the drug for two years [94,95]. PCL offers several other advantages as a drug delivery material, such as a slower degradation rate, shorter in vivo adsorbable time, and the generation of a minimal acidic environment during degradation [93]. These properties make PCL an attractive choice for controlled and efficient drug delivery.

### 4.1. Types of Drug Delivery Systems Based on PCL

#### 4.1.1. Polymeric Micelles

Polymeric micelles formed by the self-assembly of amphiphilic block copolymers of PCLs can be used as drug delivery vehicles. These amphiphilic block copolymers are composed of a hydrophilic and a hydrophobic segment which can self-assemble in aqueous media to form micelles with a hydrophobic core and a hydrophilic shell. Hydrophobic drug molecules can be encapsulated within the hydrophobic core of the micelle (Figure 9). The hydrophilic shell can provide enhanced solubility for the micelle and its cargo; it also shields against opsonization and helps the particles stealthily circulate in the bloodstream [96,97,98,99,100,101,102]. Moreover, the hydrophilic shell can be modified by attaching targeting ligands and markers that can aid the particles in selectively interacting with specific diseased cells. Polymeric micelles, ranging from 10–100 nm, remain “stealthy” by avoiding kidney-mediated excretion and bypassing spleen filtration [22,103]. Moreover, the nanosize allows them to take advantage of passive targeting through the EPR effect and is amenable to active targeting [101,104].

Tuning the properties of the PCL block copolymer by altering the pendant group present on the CL monomer is the classic approach to obtain the micelles with desired properties. Amphiphilic PEG-*b*-PCL copolymers with PCL block bearing either carboxyl (-COOH), hydroxyl (-OH), or amine (-NH_2_) group were reported by Jundi et al. [105]. The distinct functional group present at the PCL block impacted the solubility of the block, swellability, and thermodynamic stability of the micelles. Similarly, the Jayakannan group [82] synthesized amphiphilic random and block copolymers from carboxyl functionalized CL and native CL (PCL-*b*-CPCL). The CPCL block afforded hydrophilicity to the polymer and allowed these copolymers to self-assemble into nanosized micelles. Both these polymers were able to encapsulate the anticancer drug DOX and were able to internalize it into the cell nucleus. Researchers have also attached biologically active moieties as a pendant group to abet the efficiency of the encapsulated drug. For example, valproic acid pendant group-containing poly(ethylene glycol)-*b*-poly(g-2-propylpentanoate-e-caprolactone) amphiphilic copolymer released a histone deacetylase (HDAC) inhibitor, valproic acid after 72 h incubation in PBS [66].

Polysaccharides are copiously available molecules in nature. These naturally occurring molecules are, apart from inherently biodegradable and biocompatible, inexpensive, highly stable, nontoxic, hydrophilic, and possess an excellent life in the body [106,107]. These lucrative properties of polysaccharides make them ideal materials for drug delivery applications. Hemmati and Ghaemy [108] grafted PEG-*b*-PCL-*b*-PDMA triblock copolymer on Tragacanth Gum via alkyl-azide click reaction. The micelles obtained from these comb-type graft polymers rapidly released the drug in the acidic environment, making them suitable for oral drug delivery applications. Almeida et al. [109] utilized carbodiimide chemistry to obtain amphiphilic chitosan-grafted-polycaprolactone (CS-*g*-PCL) copolymers. The CMC analysis suggested that these polymers are thermodynamically stable and could load up to 5% of Paclitaxel (PTX). In another approach reported by Youssouf et al. [110], enzymatically degraded k-carrageenan to obtain oligocarragenan and subsequently grafted with PCL chains to obtain nanospherical micelles with a critical micellar concentration of 4 × 10^−5^ M. Furthermore, in vivo, and in vitro studies showed the micelles of these benign polymers are internalized into cells via endocytosis and are mainly taken up by the liver.

Although micelles obtained by linear copolymers are advantageous, they often lack high loading capacity and require higher polymer concentrations to form micelles. Polymers having non-linear architecture are sought to tackle these shortcomings [19,111]. Comparison of micelles obtained from multiarmed copolymers with their linear analogs evidenced the superior performance of these micelles as a DOX carrier. The increment in the number of arms in polymers resulted in compact micelles with low CMC values and encapsulated more drugs. Similarly, in another study, [112] star-shaped PCL-*b*-PEG polymers were prepared by ROP using porphyrin as an initiator. Apart from encapsulating high DOX concentration, porphyrin moiety in the polymer generated singlet oxygen (^1^O_2_)—a molecule playing an important role in photodynamic therapy.

#### 4.1.2. Hydrogels

Polymeric hydrogels are three-dimensional, physically or chemically crosslinked polymer networks capable of swelling by absorption of large amounts of water or biological fluid. The presence of significant water content imparts the hydrogel’s excellent biocompatibility [113,114,115,116]. PEG-*b*-PCL-*b*-PEG triblock copolymers were prepared by combining PEG-*b*-PCL block copolymer with hexamethylene diisocyanate (HMDI). The aqueous solution of this block copolymer displayed a temperature-dependent sol-gel transition. It showed sustained release of hydrophilic drug for up to 168 h and hydrophobic drug for more than 13 days in vitro [117]. The inclusion of hydrophilic PEG block in amphiphilic Agarose-PCL co-network gels enhanced equilibrium water swelling as well as mechanical properties [118]. The PEG moiety improved the miscibility of the phases of this noncytotoxic, hemocompatible, and injectable hydrogel. This phase compatibility results in higher drug encapsulation (~35%) and zeroth order release of 5-Fluorouracil was observed. In addition to this, the hydrogels prominently released the drug at the tumor site, reduced tumor growth, and reduced the toxicity of the drug in vivo [119]. In a similar approach, PTX-loaded PCL-*b*-PEG-*b*-PCL polymers were freeze-dried to obtain the powdered nanoparticles, which upon dispersion in water, gelated at 37 °C. The in vivo pharmacokinetic evaluation of the hydrogel evidenced significantly improved half-time of the drug compared to the commercial formulation Taxol^©^ [120].

#### 4.1.3. Micro/Nanospheres

Micro/nanospheres refer to an emulsion of cell or solid particles in a continuous phase. These particles possess unique advantages in drug delivery and can be prepared by various methods [121,122]. Kim and group [122] reported a one-pot method using ionic liquid to obtain PCL microspheres containing water-soluble carbon nanotube (w-CNT)-taxol complexes. Trioctylammonium chloride, an ionic liquid, was eliminated by selective extraction with 100% ethanol to obtain a gray powder. The scanning electron microscopy analysis of these microspheres showed uniformly formed homogeneous microspheres with a mean diameter of 3.24 ± 1.72 µM (Figure 10). Although the loading capacity of these microspheres was low, the in vitro studies showed a sustained release of the drug for up to 60 days. Magnetic Fe_3_O_4_ nanoparticles and DOX-embedded PCL microspheres, on the other hand, were able to encapsulate 36.7% of the drug and are also shown to have super magnetic behavior. Due to their magnetic nature, these microspheres could be directed toward the pathological site by applying a magnetic field [123].

#### 4.1.4. Drug Conjugates

In polymer-drug conjugates, a drug is chemically connected to the polymer backbone via *a* cleavable linker. This prodrug is then delivered to the pathological site by active or passive targeting, and the degradation of the linker results in drug release [96]. Since the drug is chemically bonded, the problems associated with pre-release and low loading can be circumvented. Lipid and membrane-coated PCL-(7-ethyl-10-hydroxy-camptothecin) prodrugs were developed and assessed in vitro and in vivo for their application as nanoformulations. The membrane coating improved the cell adhesion and enhanced the cellular uptake of the drug. Moreover, membrane-coated nanomedicine outperformed free and lipid-coated drugs in vivo pharmacokinetic assays [124]. In another study, the anticancer drug Paclitaxel (PTX) was bonded to PEG-*b*-PCL polymer using acetal linkage (Figure 11) to generate drug delivery carriers with on-demand release of the drug by cleavage of the bond at the lower pH [125]. Similarly, methotrexate conjugate PCL-b-PEG micelles passively target the tumor cells and show zeroth order release kinetics [126].

## 5. Strategies to Target PCL-Based Drug Delivery Carriers

At its core, the drug delivery carriers are designed to improve the drug’s efficacy by increasing its concentration at targeted sites while decreasing its toxicities at the other sites [127,128,129,130]. In the context of tumors, this is achieved either through active [131] or passive [132] targeting. Some delivery carriers may transpire complex targeting mechanisms and can be designed to release the drug upon temperature variation, pH, light, etc., to release the cargo after accumulation at the pathological site [129,131,133,134]. Such materials are called smart- or stimuli-responsive materials, and Section 6 of this article will discuss them in detail.

### 5.1. Active Targeting

In active targeting, the carrier is embellished with tumor-recognizing moieties, such as ligands or antibodies, to deliver a drug at the pathological site [135,136]. For example, Zhou et al. dedicated their efforts to developing a particular type of polymeric micelle that exhibited a response to both enzymes and redox, with active targeting capabilities [137]. This research aimed to formulate an effective mechanism for the quick release of drugs inside cells as part of an overall cancer treatment strategy. However, one of the major issues encountered was the encapsulation of the hydrophobic drug Camptothecin (CPT), which proved to be quite challenging due to its planar structure and moderate polarity [138,139]. To deal with this obstacle, CPT was chemically bonded to mPEG, using a redox-responsive linker, to create polymeric prodrugs. The attachment of Phenylboronic acid (PBA) at the terminal of the PEG segment was performed to give the nanocarriers the ability to actively target specific tumor cells, such as hepatoma carcinoma cells, that overexpress sialic acid (Figure 12) [140,141,142]. Rigorous in vitro and in vivo testing has confirmed that this nanocarrier is capable of selectively delivering therapeutic agents to target cells, thus providing highly effective cancer treatment [137].

Li et al. have developed a cutting-edge solution to deliver doxorubicin and cypate to cancer cells [143]. Here, they improved the solubility of cypate by linking it to the polymeric micelles and targeting cancer cells by binding to biotin-avidin and then being taken up through endocytosis [144,145]. Additionally, by using biodegradable photoluminescent polymers (BPLPs), they have made the micelles photoluminescent, which has helped track their movement within the body. The developed micelles have shown excellent biodegradation, photodegradability, and photocytotoxicity. These properties make them an ideal candidate for cancer treatment. In another study, Wang et al. prepared Pyridine-grafted diblock copolymer poly(caprolactone-*graft*-pyridine)-block-poly(caprolactone) [P(CL-*g*-Py)-*b*-PCL] by combining ring-opening polymerization and Cu(I) catalyzed azide-alkyne cycloaddition (CuAAC) reaction [146,147]. They were able to create core-shell nanoparticles (CSNPs) by self-assembling transferrin (Tf) and P(CL-*g*-Py)-*b*-PCL, where the PCL block helped to encapsulate DOX (with a 10% loading capacity) through hydrophobic-hydrophobic interaction [148,149,150]. The drug-loaded Tf/P(CL-*g*-Py)-*b*-PCL CSNPs exhibited effective targeting of MCF-7 cancer cells by binding to transferrin receptors (TfR) through Tf [147].

Functionalized PCL has also been studied recently for active targeting. For example, Rezaei group developed a polymer-drug conjugate that can effectively deliver PTX to target cells [151]. They used a PEG-*b*-PCL-based amphiphilic block copolymer scaffold with functional disulfide linkages, which was prepared using a controlled ring-opening polymerization method with modifications (Figure 13). PTX was chemically linked to this scaffold through a DCC-catalyzed esterification reaction. The folate-poly(ethylene glycol)-*b*-poly((α-paclitaxel-SS-caprolactone)-*co*-caprolactone) (FA-PEG-*b*-P((PTXSS-CL)-co-CL)) micelles conjugates exhibited apparent targetability to folate receptor-overexpressing HeLa cells. The resulting FA-PEG-*b*-P((PTX-SS-CL)-co-CL) micelles were stable, had a narrow size, and had a low CMC value of 5.21 mg L^−1^. The micelles showed high intracellular uptake and drug release in the tumor intracellular environment, leading to improved cytotoxicity compared to free PTX.

Another study led by de Paiva et al. on modifying the surface of polymeric micelles to improve the delivery and specificity of a new inhibitor of polynucleotide kinase/phosphatase (PNKP) to colorectal cancer (CRC) tumors [152]. To achieve this, a peptide called GE11 was used to target the epidermal growth factor receptor (EGFR) that is commonly overexpressed in about 70% of CRC tumors. The micelles were modified with GE11 and labeled with a near-infrared fluorophore, resulting in enhanced internalization by CRC cells that overexpress EGFR, as observed in vitro and a trend towards increased primary tumor homing in an orthotopic CRC xenograft in vivo. These results suggest the potential benefit of EGFR-targeted polymeric micellar formulations as monotherapeutics for aggressive and metastatic CRC tumors. However, they also highlight the need for the development of EGFR ligands with improved physiological stability and binding to EGFR [152].

### 5.2. Passive Targeting

Passive targeting primarily relies upon the distinct pathophysiological characteristic of tumors compared to normal cells. The hyperpermeable nature of the tumor allows macromolecules to internalize. At the same time, the impaired lymphatic drainage system limits their clearance- the effect is very well known as the enhanced permeability and retention (EPR) effect [132,153,154]. For example, Zhang et al. successfully developed a novel polymeric micelle that can passively target cancer cells and alter their shape in response to pH levels [155,156]. The developed micelles have been found to be highly effective in transporting anticancer drugs, such as Gambogenic acid (GNA), with an impressive loading efficiency of 83.67% and 15.20% drug loading capacity. Moreover, the bioavailability of their GNA-loaded micelles was almost four times higher, and the peak concentration (C_max_) value was close to three times higher than that of free-GNA, indicating the protective nature of the loaded micelles. By slowing down the metabolism of GNA, these micelles help to ensure that the drug remains in circulation for longer periods of time, ultimately promoting its effectiveness and improving its overall therapeutic impact [157]. 

In 2015, Tang et al. utilized PTX-functionalized PCL to passively target cancer sites [158]. The PTX-PCL conjugate displayed slightly higher drug efficacy than free PTX in MCF-7 cells after 72 hrs of incubation at concentrations between 0.31 and 5.76 mg PTX equiv·mL^−1^. In another study, Yin et al. used TBD to prepare ester-functionalized PCL block copolymers, specifically with conjugated phenylboronic acid (PBA) formate pendant group (Figure 14) [159]. Incorporating the PBA pendant group significantly increased the interaction between the polymeric carriers and DOX. Consequently, the drug-loading capacity of the micelles was increased with the use of PBA-modified PCL block copolymers with an encapsulation efficiency of over 95%. These micelles have also been observed to maintain a consistent particle size for up to a week, which makes them a reliable drug carrier.

## 6. Stimuli-Responsive PCL for Drug Release

Even though targeting disease site in the body help minimizes the side effect of anticancer drug, the ability to control when and where drugs are released is important [12,13,15,160,161]. Moreover, the therapeutic effect of a drug that is loaded in a DDS cannot be established until it is released, even if the DDS is present at the intended site. This challenge has revolutionized the design of “smart” delivery systems. The term “smart” contextually means the ability of the DDS to respond to stimuli and consequently release their payload. This has been accelerated by advanced studies on the differences between the pathology of healthy and diseased tissues. For instance, Solid tumors are reported to have lower pH, elevated reactive oxygen species (ROS) and reductive compounds such as glutathione, and ester hydrolyzing enzymes such as esterase [162,163,164,165,166]. Upon such differences, researchers have designed materials that can respond to these internal stimuli to initiate an autonomous release of the payload from the DDS into solid tumors. Materials that respond to external triggers, such as temperature, light, magnetic field, and ultrasound, have also been studied [163,167].

### 6.1. pH-Responsive PCL

Normal physiological environment, such as the blood, tightly regulates the pH around 7.4, while solid tumors have a slightly acidic pH [82,151]. Solid tumors such as breast cancer are known to have restricted blood perfusion and high glycolytic cancer cell metabolic activity, leading to decreased pH [87,168]. Hence, this low pH can be targeted to release anticancer drugs from DDS (Figure 15).

Recently, Yin et al. [169] synthesized two PCL-derived amphiphilic polymers for the delivery of paclitaxel (PTX) and doxorubicin (DOX). PTX was covalently attached to the polymer, while DOX was physically encapsulated with loading capacities of 11.6% and 12.4% for each polymer, respectively. The acid hydrolyzable ester backbone and amide bonds make this polymer pH-responsive, as the highest drug (DOX and PTX) released was recorded at a pH of 5 relative to the pH of 7.4. Niknejab et al. [151] also designed a novel folate-poly(ethylene glycol)-b-poly((a-paclitaxel-SS-caprolactone) ε-caprolactone) (FA-PEG-b-P(PTX-SS-CL)-co-CL)) amphiphilic polymer to deliver PTX. Since the PTX was attached to an acid-cleavable β-thiopropionate group, the release of PTX at low pH was determined. About 79% of PTX was released at a pH of 5.0, while only 6% release was observed at neutral pH. The effect of different chemical compositions and polymer topology (random and block copolymer) of aliphatic polyester on the pH-responsive behavior was investigated by Jayakannan et al. [82]. Three amphiphilic random copolymers made up of unsubstituted CL and carboxylic-functionalized polycaprolactone (CPCL_n_) (n represents the number of the carboxylic functionalized CL repeat unit)—were synthesized as CPCL_30_, CPCL_50_, and CPCL_70_ and loaded with DOX. It was observed that the highest DOX release of 50% was observed at a pH of 4.0 from CPCL_70_ polymer, while only 18% DOX release was observed for both CPCL_30_ and CPCL_50_ at the same pH. However, DOX release from the block copolymer, PCL_50_-b-CPCL_50_, was only 25% at the acidic condition. The DOX release from CPCL_70_ polymer was because of the larger number of ester hydrolyzable bonds from both carboxylic acid pendants and the ester polymer backbone, implying that the composition of amphiphilic polymer needs to be tuned to achieve the most pH-responsive polymer.

### 6.2. Thermo-Responsive PCL

Materials with thermal properties can be tailor-made so that their incorporation in amphiphilic polymers intended for drug delivery can retain the integrity of the DDS at a certain temperature range but release the loaded drug at another range of temperature. These materials are generally known to be thermo-responsive. Thermo-responsive polymers reported as DDS are known to undergo phase transition between a hydrated state and a dehydrated state above or below their critical solution temperature (CST) [170], as represented in Figure 16. These thermo-responsive polymers become globular in a hydrated state and coil-like once dehydrated [13,70,171]. The temperature above which thermo-responsive polymers are insoluble in aqueous solution is termed lower critical solution temperature (LCST) and is often determined as the temperature at which there is a 50% drop in the transmittance as the polymer solution is heated [12,13,160,161]. The temperature below which thermo-responsive polymer transitions from coil-to-globular shape is also known as upper critical solution temperature (UCST). The application of such a concept has found application in DDS where the DDS remains intact (drugs are not released) at physiological temperature (37 °C) but rapidly releases the loaded drug by either raising the temperature (in polymers with LCST) or reducing the temperature (in polymer with UCST) at the local area where the drug is intended to be released [12,13,160,161,170].

Over the last decade, Stefan et al. [12,13,160,161] have reported several thermo-responsive amphiphilic polymers to prepare micelles to deliver anticancer drugs. In one of the recent reports [160], linear, four-armed, and six-armed star-like amphiphilic polymers named poly(γ-benzyloxy-ε-caprolactone)-*b*-poly(γ-2-[2-(2-methoxyethoxy)ethoxy]ethoxy-ε-caprolactone) (PBCL-*b*-PMEEECL) were synthesized to determine the effect of topology on the LCST. The LCST for the linear, 4-armed, and 6-armed amphiphilic polymers was selected to be 38.9, 40.1, and 40.4 °C, respectively. The drug release profile shows a high DOX was released above the LCST (42 °C) than at 37 °C for all the polymers, with the highest release associated with the linear polymer and comparable release from both four- and six-armed polymers. However, the six-armed DOX-loaded micelles had the highest HeLa cell death. Stefan et al. [171] also demonstrated the effect of polymer composition on LCST by synthesizing four PCL-based polymers (P1,P2,P3,P4) with varying ratios of tri(ethylene glycol) (ME3) content (Figure 17A). They achieved LCST in the range of 29.9 °C to 54.2 °C for the four polymers and observed that the LCST increases as we increase the tri(ethylene glycol) content in the polymer composition (Figure 17B). This indicates that the LCST of thermo-responsive polymers can be fine-tuned by varying compositions to achieve the desired LCST for biological application.

Stefan et al. [64] also demonstrated the effect of oligo(ethylene glycol) (x) length on the LCST by synthesizing poly(γ-oligo(ethylene glycol)-ε-caprolactone-*b*-poly(γ-benzyloxy-ε-caprolactone) (PME_x_CL-b-PBnCL, x = 2, 3 and 4). The LCST for PME_2_CL-b-PBnCL PME_3_CL-b-PBnCL and PME_4_CL-b-PBnCL were 15, 41, and 59 °C respectively. An increase in the length of oligo(ethylene glycol) increases the solubility of the polymer hence increasing the LCST. In another study, they demonstrated the effect of concentration on the LCST of PCL-based homopolymers [32]. The data reveals that the LCST of the homopolymer decreases as we increase concentration. It is imperative to note that thermo-responsive polymers with LCST far beyond physiological temperature are not desirable for biological application as such elevated temperature has the propensity to damage cells.

### 6.3. Redox-Responsive PCL

Reactive oxygen species (ROS) is a group of unstable oxygen-containing radicals, molecules, and ions that react rapidly with nucleophilic compounds. ROS include singlet oxygen, hydroxyl radicals, peroxides, etc. Under normal physiological conditions, ROS are produced in low amounts, and tightly regulated to prevent cellular damage, as they have the potential to react with biomolecules like DNA, proteins, and lipids [164,166]. The amount of ROS spikes in events such as inflammation caused by microbial infections, cancer, mechanical injury, and immunological response [172]. It is documented that ROS, particularly hydrogen peroxide, and superoxide, stimulates the proliferation of several types of cancer cells, such as breast cancer cells [165]. The high metabolic activities in many tumors produce excess ROS. Cancer cells also produce reductive species, primarily reduced glutathione (GSH), to quench these ROS to minimize its lethal effect [173]. It is known that many cancer cells have elevated amounts of both ROS and GSH than in normal healthy cells. This has also become an interesting stimulus upon which DDS is designed for tumor-specific drug release [164,166].

Li et al. [174] designed an injectable hydrogel, poly(ethylene glycol)-*b*-poly(ε-caprolactone-co-1,4,8-trioxa[4.6]spiro-9-undecanone) that can be triggered by GSH to locally deliver PTX into tumor sites. In the presence of GSH, about 70% PTX was released at pH (7.4) and only 40% without GSH. A similar trend of drug release was reported by Niknejab et al. [151]. The report demonstrated that amphiphilic polymer can be designed to be both pH- and redox-responsive towards drug release. Here, they synthesized a PTX-conjugated PCL-based amphiphilic polymer to release the covalently attached PTX in the presence of GSH. About 97% and 79% of PTX were released at pH of 5.0 and 7.4 in the presence of GSH but close to 40% and 5% at pH of 5.0 and 7.4, respectively, in the absence of GSH (Figure 18). A similar observation was reported by Yin et al. [159] where three amphiphilic polymers, mPEG-*b*-P(CL_40_-co-BCCL_15_) (PBCCL), mPEG-*b*-P(CL_40_-co-CCL_15_) (PCCL) and mPEG-*b*-P(CL_40-_co-CCL_15_)-*g*-PBA (PPBA), and the effect of the presence of hydrogen peroxide (ROS) on the release of DOX from the PPBA micelle was determined. Interestingly, this polymer was both pH and redox responsive as 50% DOX was released at a pH of 5.5 and only 10% at a pH of 7.4. In the presence of radicals (hydrogen peroxide), a relatively higher amount of DOX was released at all pH, with the highest at the pH of 5.5. This indicates that amphiphilic polymers can be designed to possess multiple stimuli-responsive behaviors to release both covalently attached and encapsulated drugs.

### 6.4. Enzyme-Responsive PCL

Enzymes secreted by cancer cells that break down DDS to release anticancer drugs are available. As such, the polymer composition of the DDS exhibiting enzyme-responsiveness is invulnerable as they are protected from degradation by other enzymes during circulation and biodistribution until they reach the targeted site where the targeted enzyme initiates hydrolysis of the polymer to release the payload [175] (Figure 19). Moreover, the degraded polymer fragments can then be easily excreted out of the body to minimize any potential toxicity associated with the polymer [175].

Jayakannan et al. [176] synthesized a fully biodegradable PCL-based star-shaped polymer with three (SB_3_) and six-armed (SB_6_) block copolymers. The six-armed polymer, SB_6_, had the highest drug loading capacity of 15.2%. At acidic pH, both polymers had about 30% DOX released. However, in the presence of esterase, an enzyme known to hydrolyze ester bonds, there was a drastic release of DOX, with the highest release of about 90% from both polymers. Similarly, in another work, they studied the enzyme-responsive behavior based on polymer topology [81]. Here, two highly luminescent triblock copolymers, A-B-A comprising biodegradable PCL and B is cationic oligo-phenylenevinylene (OPV), were synthesized. The micelles obtained from BPCL_60_ and BPCL_100_ reported a DOX loading capacity of 6.5% and 6.3%, respectively. It was observed that 80% of DOX was released in the presence of esterase, while only 15% was observed in its absence. Jayakannan and co-coworkers [177] also synthesized perylenebisimide (PBI) tagged PCL, PBI-CPCL_40_, that self-assembles in aqueous media to form red-fluorescent micelles and nanofiber in organic solvents. The red-fluorescent micelles were <150 nm in diameter with fluorescent quantum yield (ϕ) from 0.25 to 0.30, which is desirable for bio-imaging. The formation of multimodal from monomodal distribution in the presence of esterase indicates the cleavage of the ester backbone of the amphiphilic polymers. This was also confirmed through size exclusion chromatography, where the molecular weight of the polymer decreased with increasing incubation time.

## 7. Conclusions and Future Outlook

PCL has emerged as a valuable tool in biomedical research due to its biodegradable and biocompatible properties. Its synthetic versatility allows for tuning of its properties to suit specific applications. One significant challenge still faced by researchers in this field is the ROP of multi-functional CL monomers. While some progress has been made, including the ROP of ester and amide functionalized CL monomers, there is still a need to develop an efficient catalytic system for the ROP of functional CL monomers. Ongoing research aims to further design and attach novel functionalities to PCL-based systems to achieve precise control over drug delivery and enhance therapeutic outcomes. Integration of PCL with other biopolymers, such as polysaccharides, opens up new possibilities with tailored properties and functionalities. Continued advancements in the synthesis, functionalization techniques, and a deeper understanding of the drug release mechanism are expected to drive the development of effective PCL-based drug delivery systems.

## Figures and Tables

**Figure 1 pharmaceutics-15-01977-f001:**
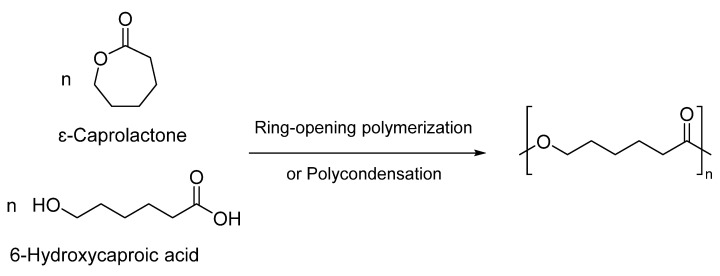
Synthesis of poly(ε-caprolactone) through ROP or polycondensation [24].

**Figure 2 pharmaceutics-15-01977-f002:**
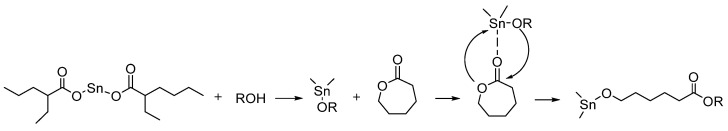
Coordination-insertion mechanism of CL polymerization with Sn(Oct)_2_ as a catalyst [15].

**Figure 3 pharmaceutics-15-01977-f003:**
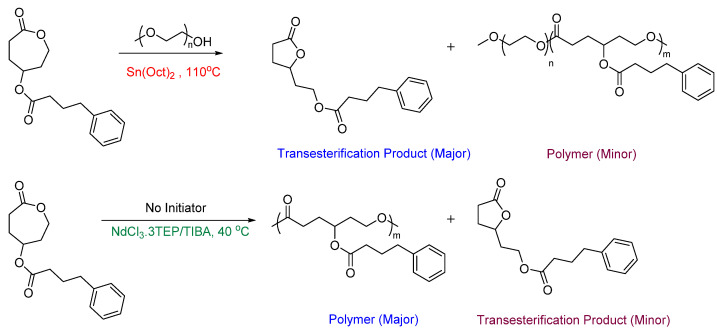
Polymerization of γ-4-phenylbutyrate-CL monomer using two different catalytic systems [36].

**Figure 4 pharmaceutics-15-01977-f004:**
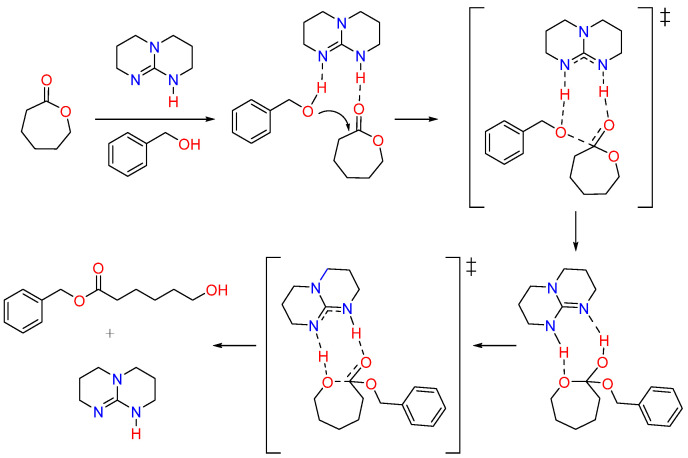
Mechanism of polymerization of CL monomer using TBD catalyst [50]. ‡ indicates the reaction intermediate formed during the reaction.

**Figure 6 pharmaceutics-15-01977-f006:**
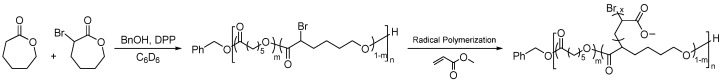
Grafting PMA on PCL using bromo-PCL as a macroinitiator [76].

**Figure 7 pharmaceutics-15-01977-f007:**
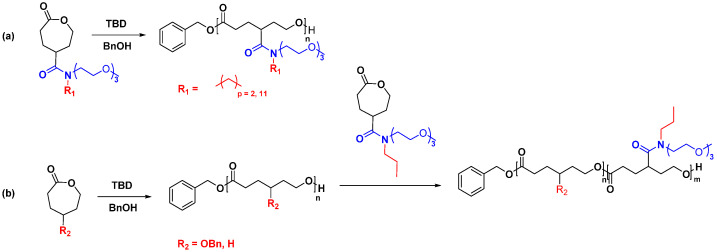
Synthesis of (**a**) amphiphilic homopolymer, (**b**) amphiphilic block copolymer based on amide functionalized PCL [32,70].

**Figure 8 pharmaceutics-15-01977-f008:**
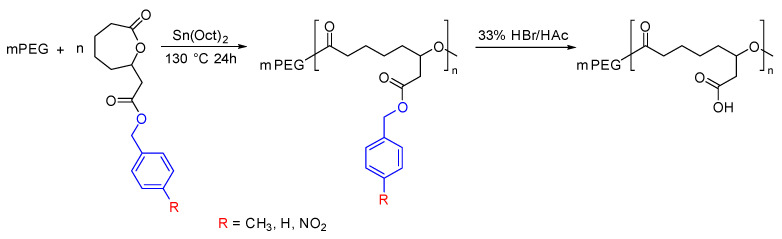
Synthesis of carboxylic acid functionalized PCL by varying the protecting group [73].

**Figure 9 pharmaceutics-15-01977-f009:**
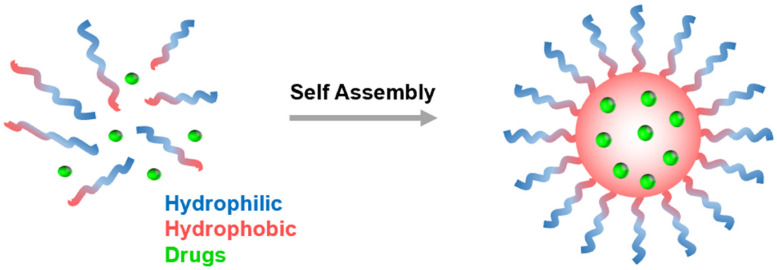
Polymeric micelles formed from the self-assembly of amphiphilic block copolymers in aqueous solution.

**Figure 10 pharmaceutics-15-01977-f010:**
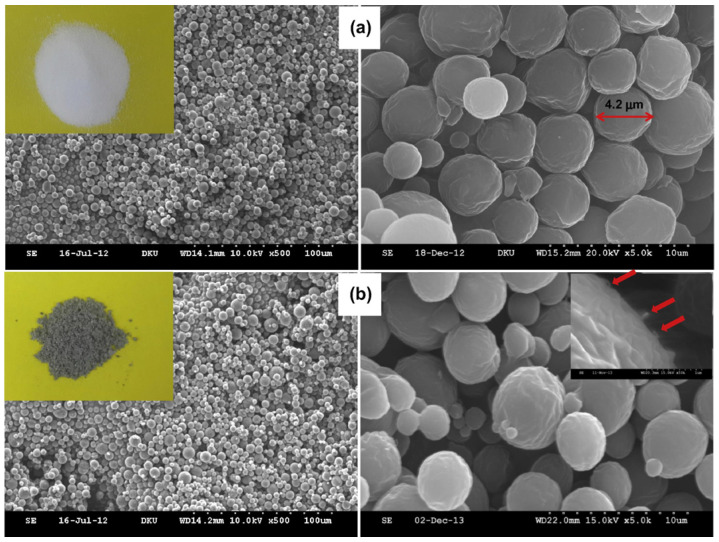
Scanning electron micrographs of (**a**) pure PCL microspheres and (**b**) w-CNT/taxol-containing PCL microspheres (PCTx-1). In the insets: (i) photos of pure PCL microspheres (the left upper inset, white color) and of *w*-CNT/taxol-containing PCL microspheres (the left lower inset, gray); (ii) the red arrows indicate projected portions of *w*-CNTs (the right lower inset) [122] (Printed with permission from Elsevier©).

**Figure 11 pharmaceutics-15-01977-f011:**
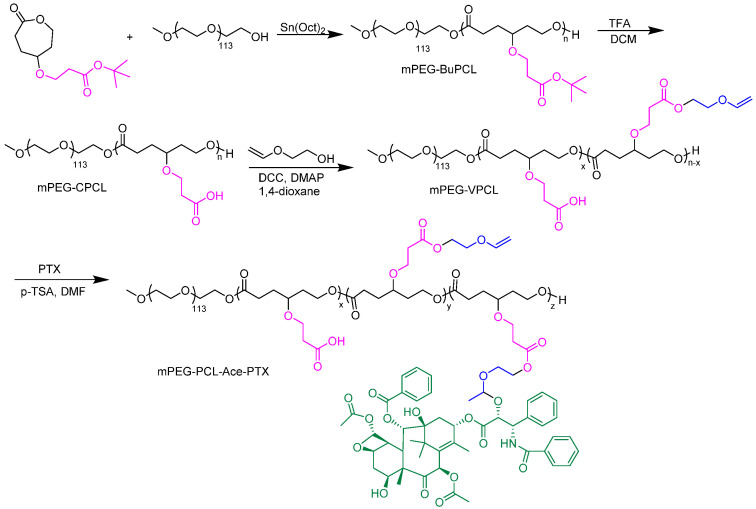
Synthesis of paclitaxel conjugated PEG-b-PCL polymers [125].

**Figure 12 pharmaceutics-15-01977-f012:**
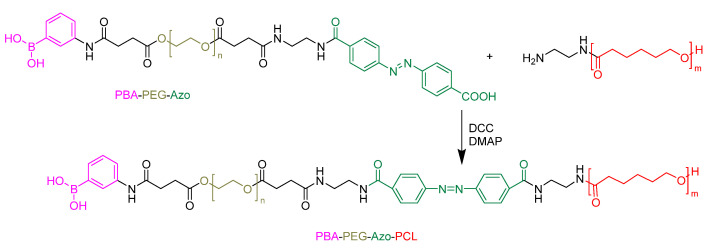
Phenyl boronic acid conjugated to PCL-based micelles for active targeting [137].

**Figure 13 pharmaceutics-15-01977-f013:**
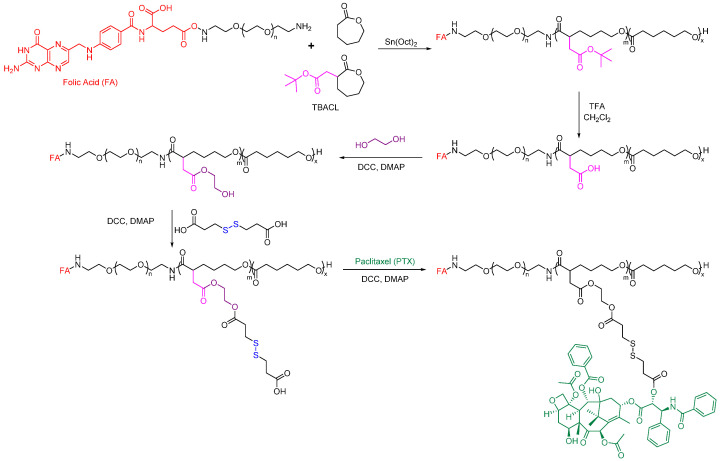
Folate-conjugated PCL-based polymeric micelles for the delivery of Paclitaxel [151].

**Figure 14 pharmaceutics-15-01977-f014:**
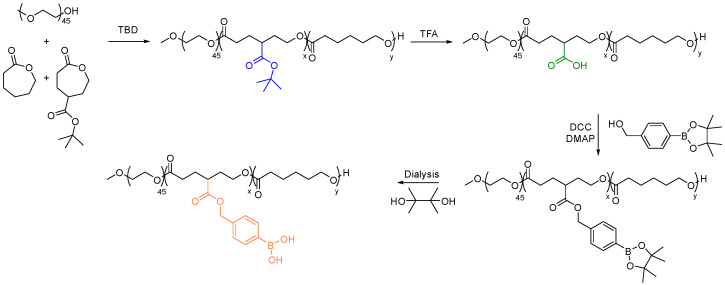
Synthesis of PBA substituted PCL for enhanced efficiency of DOX delivery [159].

**Figure 15 pharmaceutics-15-01977-f015:**
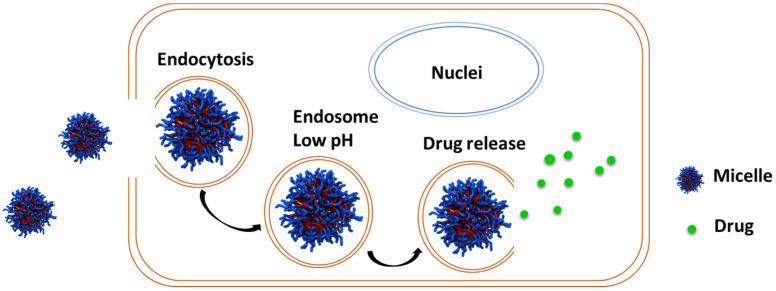
Schematic representation of drug release from micelles when present in lower pH.

**Figure 16 pharmaceutics-15-01977-f016:**
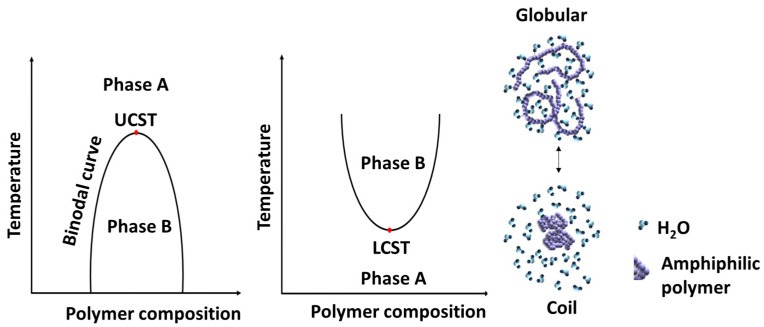
Schematic representation of the phase transition of thermo-responsive polymer in aqueous solution [170].

**Figure 17 pharmaceutics-15-01977-f017:**
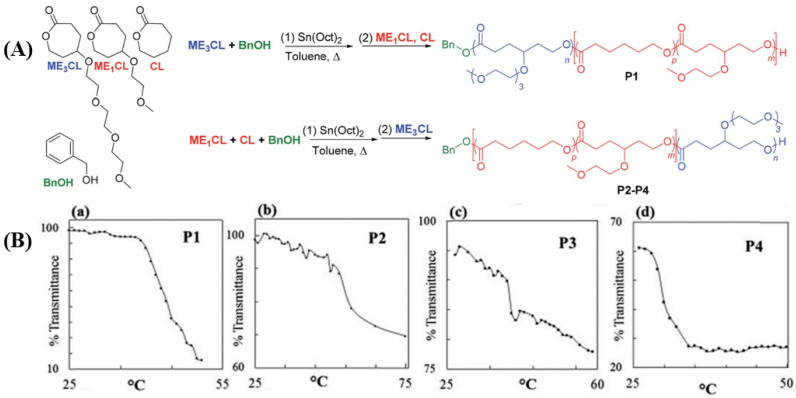
(**A**) Synthesis of thermo-responsive PCL with varying content of ME_3_ unit. (**B**) The LCST of the polymer decreases as we go from P1 to P4 due to the decrease in ME_3_ content [171] (Printed with permission from the Royal Society of Chemistry).

**Figure 18 pharmaceutics-15-01977-f018:**
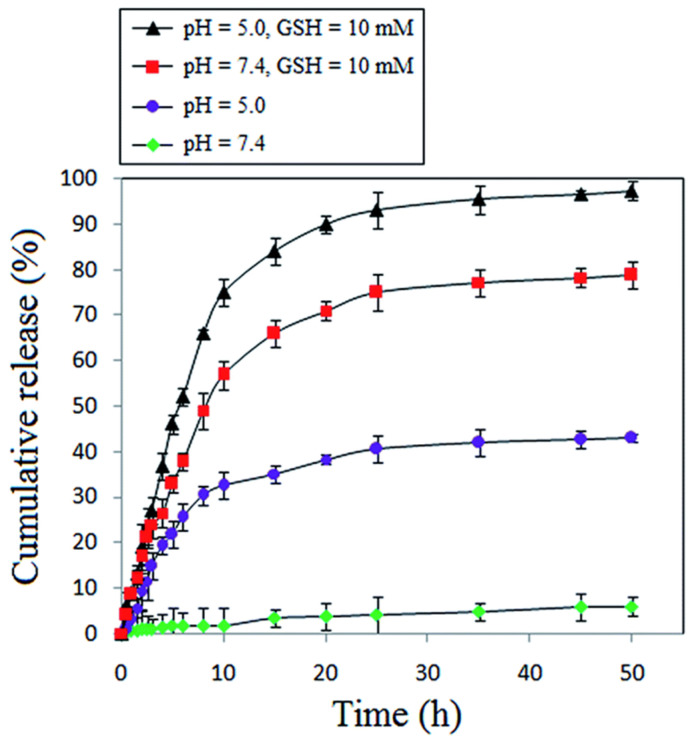
Cumulative release of PTX in the presence/absence of GSH at different pH [151] (Printed with permission from the Royal Society of Chemistry).

**Figure 19 pharmaceutics-15-01977-f019:**
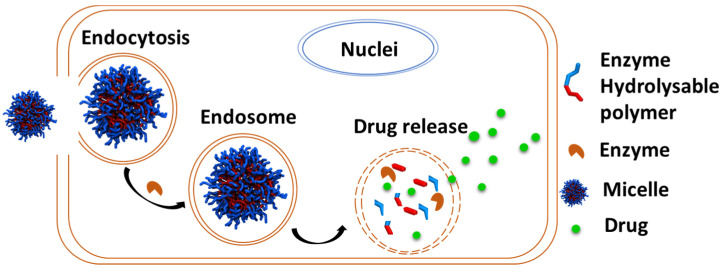
Schematic illustration of enzyme-responsive polymer used for drug delivery.

## Data Availability

Data sharing not applicable.

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
