# Peer review of "Recent Advances in Polycaprolactones for Anticancer Drug Delivery"

_pharmaceutics, 2023, doi:10.3390/pharmaceutics15071977_

Round 1

Reviewer 1 Report

A review article by Bhadran et al analyzes recent publications on the application of polycaprolactones (PCL) as drug delivery vehicles. The review consists of two logical blocks, namely, chemical (first part of the manuscript) and biomedical (second part). The "chemical" part of the review describes in detail the methods for the synthesis and modification of PCL. To me, this part is quite informative and comprehensive, but a bit lengthy and overloaded by information with no relation to the delivery, such as catalysts to be used etc. However, I do not insist on my point of view and have no substantial questions to it.

            The second “biomedical” part of the paper deals with the PCL-mediated delivery of drugs.  This part is also expertly written, but looks a bit more fragmented, which is obviously due to the large amount and variety of data on this issue. I have a few comments on it to be addressed.

1. In fact, all the publications analyzed in the review are related to the delivery of cancer therapy molecules (mostly hydrophobic), such as DOX, PTX, GNA. At the same time, there are a number of works on the modified PCL-mediated delivery of siRNAs and mRNAs. For instance, “Delivery of siRNA to the brain using a combination of nose-to-brain delivery and cell-penetrating peptide-modified nano-micelles”, DOI: 10.1016/j.biomaterials.2013.08.036, or “mRNA delivery using non-viral PCL nanoparticles”, DOI: 10.1039/c4bm00242c, etc. (please see PubMed). Certainly, these important data are the drug delivery data, too. If the authors do not want to include them, I would suggest to modify the title something like “Recent Advances in Polycaprolactones for the anti-cancer Drug Delivery”, to specify the point of the review.

2. Unit 5.2 “Passive targeting”. Why do the authors include reference 158 here? I’m sure that interaction of transferrin to the transferrin receptor is a prime example of active targeting, similar to the all references listed in 5.1.

3. Unit 6.1. “pH-Responsive PCL”. The authors write (lines 491-494) “the tumor cell…. export protons and lactic acid out of the cell thereby creating an acidic extracellular microenvironment [166]. Hence, this low pH can be targeted to release anti-cancer drugs from DDS (Figure 17)”. Clearly, this is nonsense. Anti-cancer drugs should be delivered inside the tumor cell. If extracellular pH played a role there, we would see a premature drug release and no therapeutic effect. Instead, Fig. 17 correctly shows intracellular release as a result of low pH in the endosome (endosomal escape). So, please re-write the above sentences.  

4. I suppose, there is a mistake on the same Fig. 17. Definitely, the PCL nanoparticles – delivered drug is released before the late endosome fuses with the lysosome. Lysosome rupture, as is shown there, would be a total catastrophe for the cell. Maybe, in terms of oncology it would be rather good, but I afraid it does not happen in reality. So, I think the word “lysosome” should be replaced by the “endosome”.  

 5. Unit 6.2. “Thermo-Responsive PCL”. In the section, a very wide range of temperatures is mentioned, in sum, 15 -  59 °C. Of course, most of them are far beyond the physiological conditions. So, though interesting, those data are scarcely applicable for the delivery. I think, it is necessary to set this down in the text.

After paying attention to the above comments, the paper can be re-considered for the publication.    

Reviewer 2 Report

Dear Editor:

In this review, the authors summarized the synthesis method of PCL, and the application of functionalized PCL. At the same time, the related dosage forms and targeting strategies for PCL drug delivery system are summarized. It is of great reference value for the application of PCL. There are several suggestions as follows:

1.Detailed and complete description of the reason why PCL can be used for drug delivery needs to be provided in the introduction.

2.Please indicate the source when quoting the document pictures.

3.The logic of the topic is insufficient.

4.In Section 3, descriptions need to be classified.

5.The reference is not close to the recent research progress. The authors need to cite newer literature, such as ACS Appl Mater Interfaces 2022;14:11092-11103, WIREs Nanomedicine and Nanobiotechnology 2020;13:e1670, Regenerative Biomaterials, 2023, 10, rbad048.  

6.The targeting strategies of PCL are not enough.

7.In Section 6.3, please separate ROS-responsive PCL and GSH-responsive PCL.

Reviewer 3 Report

Accepted as it is.

Some changes required.

Author Response

no comments

Reviewer 4 Report

1-    In Fig 2 and 11 please mention  the reference 

2-    If possible, please increase the resolution of Figure 18 and include the reference.

3-    Either Figure 5 is misplaced, or the figure numbers are out of order.

4-    Figure 10 is not mentioned in the text.

5-    Either Figure 15 is misplaced, or the figure numbers are out of order.

6-    There is 23% text similarity for your manuscript. If you reduce it a bit, the quality of the article will improve. The similarity file attached

7-    Use these articles to improve the manuscript. https://doi.org/10.1080/17425247.2022.2041598 , https://doi.org/10.1016/j.chemphyslip.2022.105179,

Round 2

Reviewer 1 Report

I thank the authors for their careful work on the manuscript. I'm happy with the authors' response, and, I believe, now the paper can go further to the publishing.